# Alteration of Gene Expression After Entecavir and Pegylated Interferon Therapy in HBV-Infected Chimeric Mouse Liver

**DOI:** 10.3390/v16111743

**Published:** 2024-11-06

**Authors:** Huarui Bao, Serami Murakami, Masataka Tsuge, Takuro Uchida, Shinsuke Uchikawa, Hatsue Fujino, Atsushi Ono, Eisuke Murakami, Tomokazu Kawaoka, Daiki Miki, Clair Nelson Hayes, Shiro Oka

**Affiliations:** 1Department of Gastroenterology, Graduate School of Biomedical and Health Sciences, Hiroshima University, Hiroshima 734-8551, Japan; d215797@hiroshima-u.ac.jp (H.B.); serami@hiroshima-u.ac.jp (S.M.); shinuchi@hiroshima-u.ac.jp (S.U.); fujino920@hiroshima-u.ac.jp (H.F.); atsushi-o@hiroshima-u.ac.jp (A.O.); emusuke@hiroshima-u.ac.jp (E.M.); kawaokatomo@hiroshima-u.ac.jp (T.K.); daikimiki@hiroshima-u.ac.jp (D.M.); nelsonhayes@hiroshima-u.ac.jp (C.N.H.); oka4683@hiroshima-u.ac.jp (S.O.); 2Liver Center, Hiroshima University Hospital, Hiroshima 734-8551, Japan; 3Division of Travel Medicine and Health, Research Center for GLOBAL and LOCAL Infectious Diseases, Oita University, Oita 879-5593, Japan; tuchida@oita-u.ac.jp; 4Department of Gastroenterology, Faculty of Medicine, Oita University, Oita 879-5593, Japan

**Keywords:** hepatitis B virus, antiviral therapy, human hepatocyte, gene expression, next-generation sequencing

## Abstract

Cross-sectional analyses using liver tissue from chronic hepatitis B patients make it difficult to exclude the influence of host immune responses. In this study, we performed next-generation sequencing using the livers of hepatitis B virus (HBV)-infected uPA/SCID mice with humanized livers before and after antiviral therapy (AVT) with entecavir and pegylated interferon, and then performed a comparative transcriptome analysis of gene expression alteration. After HBV infection, the expression of genes involved in multiple pathways was significantly altered in the HBV-infected livers. After AVT, the levels of 37 out of 89 genes downregulated by HBV infection were restored, and 54 of 157 genes upregulated by HBV infection were suppressed. Interestingly, genes associated with hypoxia and KRAS signaling were included among the 54 genes upregulated by HBV infection and downregulated by AVT. Several genes associated with cell growth or carcinogenesis via hypoxia and KRAS signaling were significantly downregulated by AVT, with a potential application for the suppression of hepato-carcinogenesis.

## 1. Introduction

The hepatitis B virus (HBV) is a member of the *Hepadnaviridae* family and contains a 3.2 kb partially double-stranded circular DNA genome within the viral particle. A universal vaccination program for preventing hepatitis B virus (HBV) infection has been promoted worldwide; however, an estimated 1.5 million people become newly infected with HBV each year, while 296 million people remain chronically infected with HBV [1]. To prevent the development of advanced liver dysfunctions, such as liver cirrhosis, hepatocellular carcinoma, and liver failure, antiviral therapies using pegylated interferon (PEG-IFN) and/or nucleotide/nucleoside analogs (NAs) are conducted to treat chronic hepatitis and liver cirrhosis caused by chronic HBV infection [2,3,4]. Once HBV infects human hepatocytes, the viral genome is transported into the nucleus, where it forms a covalently closed circular DNA (cccDNA) mini-chromosome, making it difficult to eradicate HBV from hepatocytes through the current antiviral therapies. Therefore, the current guidelines for managing chronic hepatitis B and liver cirrhosis recommend long-term treatment using NAs [5,6,7,8].

Antiviral therapies, especially NA therapy, suppress serum HBV DNA levels below the limit of detection and maintain serum ALT levels within the normal range. However, even when hepatitis is suppressed via the long-term inhibition of HBV replication, the incidence of hepatocellular carcinoma (HCC) is reduced but not eliminated [9]. Because HBV mRNA continues to be transcribed from cccDNA and HBV-related proteins continue to be produced during antiviral therapy (AVT), the alterations of gene expression due to HBV infection persist in hepatocytes during AVT, and signal pathways associated with hepato-carcinogenesis might remain activated. However, this point has not been fully analyzed, and it is not clear whether the alteration of gene expression profiles in HBV-infected hepatocytes can recover to pre-infection levels following AVT.

Previously, we succeeded in constructing a mouse model that can be continuously infected with HBV using human hepatocyte chimeric mice [10]. Since the chimeric mice were generated from severe combined immunodeficiency (SCID) mice, they remain in a severely immunodeficient condition, and most mouse hepatocytes can be replaced with transplanted human primary hepatocytes with low rejection [10,11]. This chimeric mouse model can be used to examine the response of human hepatocytes against infected viruses without the confounding effects of the host immune response [12,13,14]. We have demonstrated the attenuation of IFN responsiveness and the induction of inflammatory cytokines and chemokines or histone methyltransferases in human hepatocytes after HBV infection by cDNA microarray and next-generation sequencing [15,16,17]. On the other hand, we have also evaluated the antiviral effects of PEG-IFN and/or NAs using this chimeric mouse model [18,19,20], and clarified that high-dose PEG-IFN plus entecavir therapy could reduce serum HBV DNA to undetectable levels [18].

In this study, we performed a gene expression analysis by next-generation sequencing using human hepatocytes obtained from HBV-infected humanized mice that were treated with or without high-dose PEG-IFN plus entecavir therapy. Gene expression profiles were compared with those of uninfected humanized mice to assess the recovery of gene expression profiles by AVT. We identified several genes associated with cell growth or carcinogenesis via hypoxia and KRAS signaling that were significantly downregulated by AVT, with a potential application for the suppression of hepato-carcinogenesis.

## 2. Materials and Methods

### 2.1. Human Serum Samples

Serum samples were obtained from HBV carriers after obtaining written informed consent. The inoculum was obtained from an HBV carrier positive for hepatitis B surface antigen (HBsAg) and hepatitis B e antigen (HBeAg) with high-level viremia (HBV genotype C2, serum HBV DNA 9.2 log copies/mL) and used to generate HBV-infected mice. The sequence of the inoculated HBV clone was registered in the National Center for Biotechnology Information (NCBI) database (Accession number: MH887433). The experimental protocol meets the ethical guidelines of the Declaration of Helsinki and was approved by the Hiroshima University Ethical Committee (Approval ID: D08-9).

### 2.2. Human Hepatocyte Chimeric Mouse Experiments

The preparation of uPA^+/+^/SCID^+/+^ mice and transplantation of human hepatocytes were performed as described previously [11]. All mice were transplanted with frozen human hepatocytes obtained from the same donor, and the mice in which more than 90% of the liver tissue had been replaced by transplanted human hepatocytes based on human albumin levels in the mouse serum were used in this study. All the animal protocols described in this study were performed in accordance with the guidelines of the local committee for animal experiments at Hiroshima University, and all the animals received humane care (Approval ID: A22-173-2).

For the next-generation sequencing gene expression analysis, 11 chimeric mice with transplanted human hepatocytes (lot: WF) were prepared and divided into three experimental groups (Figure 1a). Group 1, which served as a control, contained five mice that were not infected with HBV. The remaining 6 mice were inoculated with human serum #1 containing 6.0 × 10^6^ copies of HBV via the mouse tail vein. After the serum HBV DNA levels plateaued, 3 of the 6 mice (Group 2) were sacrificed without AVT, and the mouse livers were explanted. The final 3 mice (Group 3) were treated with entecavir (0.5 mg/kg, orally, daily) plus pegylated interferon α2a (300 μg/kg, subcutaneous injection, twice weekly) until HBV DNA became undetectable. Then, the mice in Group 3 were sacrificed and the mouse livers were explanted.

Infection, the extraction of serum samples, and the sacrifice of the animals were performed under ether anesthesia. Human serum albumin in mouse serum was measured with a Human Albumin ELISA Quantitation kit (Bethyl Laboratories Inc., Montgomery, TX, USA) according to the manufacturer’s instructions. The serum samples obtained from the mice were aliquoted and stored at −80 °C until use.

### 2.3. Analysis of HBV Markers

For the quantitative analysis of serum HBV DNA, 10 µL of mouse serum was used. Serum HBV DNA was quantified by real-time PCR using the TaqMan PCR System (Roche Diagnostics, Tokyo, Japan). The lower quantitation limit of this assay is 4.4 log copies/mL.

For the quantitative analysis of intrahepatic HBV RNA, total RNA was extracted from chimeric mouse livers by NucleoSpin RNA II (MACHEREY-NAGEL GmbH & Co. KG, Düren, Germany). After the total RNA was reverse-transcribed to complementary DNA (cDNA) using reverse-transcriptase ReverTra Ace (Toyobo, Tokyo, Japan) and random primer in accordance with the instructions supplied by the manufacturer, intrahepatic HBV RNA was measured by real-time PCR with the following protocol: The generated cDNA was quantified by real-time PCR using the 7300 Real-Time PCR System (Applied Biosystems, Foster City, CA, USA), with Glyceraldehyde 3 phosphate dehydrogenase (GAPDH) expression used as a control. Amplification was performed in a 25 μL reaction mixture containing 12.5 μL of SYBR Green PCR Master Mix (Applied Biosystems, Foster City, CA, USA), 5 pmol each of forward and reverse primer, and 1 μL of cDNA solution. After incubation for 2 min at 50 °C, the sample was denatured for 10 min at 95 °C, followed by a PCR cycling program consisting of 40 cycles of 15 s at 95 °C, 30 s at 55 °C, and 60 s at 60 °C. The primer sequences were as follows: GAPDH forward 5′-ACAACAGCCTCAAGATCATCAG-3′ and reverse 5′-GGTCCACCACTGACACGTTG-3′; HBV RNA forward 5′-TTTGGGGCATGGACATTGAC-3′ and reverse 5′-GGTGAACAATGGTCCGGAGAC-3′.

### 2.4. Dissection of Mouse Livers and Total RNA Extraction from Human Hepatocytes in the Mouse Livers

All 11 chimeric mice were sacrificed by anesthesia with diethyl ether. Human hepatocytes from the mouse livers were finely dissected and stored in liquid nitrogen after submergence in RNA later^®^ solution (Applied Biosystems, Foster City, CA, USA). The total RNA was extracted by NucleoSpin RNA II (MACHEREY-NAGEL GmbH & Co. KG, Düren, Germany). The extracted RNA quality was verified by Absorption analysis, Electrophoresis, and Bioanalyzer (Agilent Technologies, Palo Alto, CA, USA).

### 2.5. Construction of Sequence Library and Gene Expression Analysis

A sequence library was constructed by the SMART method provided by Takara Bio USA (Mountain View, CA, USA). Briefly, both the 5′ and 3′ ends of the total RNA were linked with RNA adaptors using SMART-Seq v4 Ultra Low Input RNA Kit (Takara Bio, CA, USA) according to the manufacturer’s instructions. Then, reverse transcription was performed using a specific primer that recognizes the adaptors fragmented at both 5′ and 3′ end of RNAs, and subsequently, polymerase chain reaction was performed using single-stranded cDNA templates.

Next-generation sequencing was performed using the NovaSeq6000 system (Illumina, Tokyo, Japan). Data analysis was performed by the Expression Miner software version 1.3.13 provided by Takara Bio (Tokyo, Japan) after the exclusion of the sequences derived from mouse tissues. A comparison of the gene expression levels between the two groups was performed using Student’s *t*-test (Q < 0.05). All DEGs met the following criteria: |log2(fold change)| ≥ 1 and *p* value ≤ 0.01.

### 2.6. Pathway Analysis

Analysis of canonical pathways was performed using the PANTHER software (http://www.pantherdb.org/ accessed on 30 October 2024).

### 2.7. Statistical Analysis

Pairwise differences between the groups were examined for statistical significance using Student’s *t*-test. All *p* values less than 0.05 by a two-tailed test were considered significant.

## 3. Results

### 3.1. Construction of HBV-Infected Chimeric Mice and Antiviral Treatment to HBV Infection

After inoculating with human serum containing HBV, the serum HBV DNA levels in the mice were measured every 1 or 2 weeks. All six mice developed viremia with a high viral load by 12 weeks after inoculation (more than 9 Log copies/mL of HBV DNA). After the serum HBV DNA plateaued, antiviral treatment was started using both entecavir and pegylated interferon-α2a in three of the mice. During the antiviral treatment, the serum HBV DNA titer decreased to undetectable levels within 10 weeks (Figure 1b). The intrahepatic HBV RNA level after the antiviral treatment was significantly reduced compared to that before the antiviral treatment (Figure 1c).

### 3.2. Comparison of Intrahepatic Gene Expression Between HBV-Infected and Uninfected Mice

To clarify the influence of HBV infection on intrahepatic gene expression, comprehensive RNA sequencing analysis was performed using the total RNA obtained from the HBV-infected or uninfected mouse livers. Gene expression profiles were compared using the Expression Miner software (Group 1 vs. 2). After HBV infection, 157 genes were upregulated, and 89 genes were downregulated (Figure 2). To identify the influence of HBV infection on intracellular signals, a pathway analysis was performed on this set of differentially regulated genes using the PANTHER software. After HBV infection, the expression of the genes involved in the gonadotropin-releasing hormone receptor pathway, CCKR signaling, integrin signaling, the inflammation pathway, and T cell activation were significantly altered in the HBV-infected livers (Table 1).

### 3.3. Recovering of Gene Expression Profiles by AVT

It has not been clarified whether gene expression profiles altered by HBV infection recover following antiviral treatment. To clarify this point, we compared the gene expression profiles between Group 2 (HBV infection) and Group 3 (after antiviral treatment). After AVT using entecavir and pegylated interferon, the levels of 37 out of 89 genes that had been downregulated by HBV infection were restored, and 54 of 157 genes upregulated by HBV infection were suppressed. Interestingly, the genes associated with hypoxia and KRAS signaling were included among the 91 genes for which the gene expression levels recovered to the pre-infection levels following AVT (Figure 3 and Figure 4).

## 4. Discussion

We have previously demonstrated that the human hepatocyte chimeric mouse model can support HBV infection for more than 24 weeks, and we have shown that this mouse model is suitable for analyzing the effects of viral infection and drug responses under immunodeficient conditions [12]. Furthermore, we have performed next-generation sequencing using this mouse model with HBV infection, and have compared gene expression profiles between mouse livers with and without HBV genotype C infection [16,21]. In previous studies, several pathways associated with inflammation or carcinogenesis were found to be implicated in HBV genotype C infection [21]. In the present study, we performed a similar gene expression analysis using human hepatocyte chimeric mice in which the mouse livers were transplanted with the same clone of human hepatocytes but were inoculated with a different HBV genotype C clone. As shown in Table 1, similar pathways to those associated with HBV genotype C infection, such as the p53 pathway, gonadotropin-releasing hormone receptor pathway, integrin signaling pathway, and inflammation mediated by chemokine and cytokine signaling pathway, were identified as the top 10 signaling pathways regulated by HBV infection, suggesting a high reproducibility with respect to the earlier studies.

On the other hand, it has not been clarified whether gene expression in human hepatocytes recovers after HBV reduction by AVT. Previously, we reported that a high-dose combination AVT using entecavir and pegylated interferon could reduce serum HBV DNA to undetectable levels [18]. In the present study, the HBV-infected chimeric mice were treated with entecavir and pegylated interferon until the serum HBV DNA fell to unmeasurable levels, and then, next-generation sequencing was performed using the human hepatocytes obtained from these mice. As shown in Figure 2, the expression levels of many genes recovered to levels similar to that of the mice without the HBV infection. When we checked the top 10 pathways associated with recovery after entecavir and pegylated interferon combination therapy, several pathways associated with carcinogenesis, such as hypoxia, G2M checkpoint, and TNF-α signaling via NF-κB, were identified.

In human hepatocytes experiencing hypoxia, the activation of pathways associated with G2/M checkpoint and TNF-α signaling via NF-κB are considered to accelerate hepato-carcinogenesis. Hypoxia is a fundamental feature of solid tumors [22]. A hypoxic tumor microenvironment is caused by increased oxygen consumption due to proliferation and reduced oxygen delivery [23]. Hypoxia is driven by HIFs (hypoxia-inducible factors) and involves the activation of signal pathways and transcriptional regulation of hundreds of genes [24,25]. In the present study, the expression of HIF-1α target genes, such as metallothionein 2A (MT2A), insulin-like growth factor-binding protein (IGFBP1), dual-specificity phosphatase 1 (DUSP1), metallothionein 1E (MT1E), angiopoietin-like 4 (ANGPL4), DNA damage-inducible transcript 4 (DDIT4), and MAX interactor 1 (MXI1), was upregulated after the HBV infection and downregulated by AVT (Figure 3 and 4). ANGPL4 and IGFBP1 are known as the regulators of glucose homeostasis, lipid metabolism, and insulin sensitivity, and are also known to act as an apoptosis survival factor in hypoxia [25,26]. DDIT4 is induced by endoplasmic reticulum stress and inflammation and plays an important role in protective responses to cellular stress, including responses to hypoxia and DNA damage [27]. According to our previous study, endoplasmic reticulum stress is induced by the accumulation of HBs proteins in the endoplasmic reticulum [16]. Thus, the upregulation of DDIT4 might help to resolve cellular stress. Metallothionein family members, such as MT2A and MT1E, which serve as anti-oxidants by protecting against hydroxyl free radicals [28], might be induced to control homeostasis in hepatocytes. MXI1 is a transcriptional repressor that negatively regulates the function of MYC, the accumulation of which activates cell cycle progression and cellular transformation in several cancers [29]. Accordingly, HBV infection could induce intracellular oxidative stress in human hepatocytes and establish a pro-carcinogenic environment. Meanwhile, human hepatocytes attempt to maintain intracellular homeostasis. Furthermore, AVT led to the recovery and upregulation of these genes, indicating that AVT could reduce intracellular oxidative and endoplasmic reticulum stress.

There are several limitations in the present study. The sample size in our study was relatively small, and the serum from the enrolled HBV donor patient was infected only with HBV genotype C. As individual differences cannot be ruled out, the results should be confirmed in the future using a larger sample size. While serum HBV DNA reflects the level of HBV in infected mice, serum HBV RNA, cccDNA, and HBcrAg are also correlated with intrahepatic cccDNA levels in human patients. Liver tissue extracted from human hepatocyte chimeric mice also unavoidably includes a small proportion of mouse-derived cells, including interstitial cells, bile duct cells, and vascular cells. To avoid contamination with mouse tissue, only human hepatocyte chimeric mice in which mouse liver tissue had been more than 90% replaced with human hepatocytes were used in this study. Although sequences that were clearly derived from mouse mRNA were excluded, the mapping of next-generation sequencing data may be influenced by cross-hybridization with mouse mRNA in these liver samples because the human and mouse genomes have high homology. However, since a previous study demonstrated the feasibility of gene expression analysis in chimeric mice using a functional genomics approach [30], human hepatocyte chimeric mice without HBV infection were established as negative controls, and gene expression profiles were compared using fold changes from negative controls. Finally, the measurable HBV DNA level was higher than that normally encountered in clinical situations. When more than 1.0 LogIU/mL (1.8 Log copies/mL) of HBV DNA is present in the serum, serum HBV DNA level can be measured using 500 μL of serum by the TaqMan PCR System (Roche Diagnostics). However, only 1mL of blood can be obtained from human hepatocyte chimeric mice at sacrifice, and less than 100 μL of blood can be obtained at regular blood collection intervals. In the present study, we were able to use 10 μL of mouse sera for HBV DNA measurement, but viral titers of less than 4.4 Log copies/mL of HBV DNA are considered below the measurable level. Therefore, a potential concern is that the serum HBV DNA level might not be reduced sufficiently. To avoid this difficulty, we sacrificed the mice more than 2 weeks after the serum HBV DNA levels became unmeasurable (Figure 1a). Furthermore, we also measured intrahepatic HBV RNA in order to confirm the suppression of HBV replication. As shown in Figure 1b, the intrahepatic HBV RNA level after the antiviral treatment was significantly reduced compared to that before the antiviral treatment. We conclude that the HBV replication in human hepatocytes was strongly suppressed and that we are able to analyze the impact of HBV suppression on intracellular gene expression.

## 5. Conclusions

In conclusion, we performed intrahepatic gene expression analysis using HBV-infected human hepatocyte chimeric mice and demonstrated that intrahepatic carcinogenic pathways stimulated and activated by HBV infection could be suppressed by antiviral treatment. These results support the view that maintaining low serum HBV DNA levels by inhibiting HBV replication with antiviral treatment can reduce the risk of hepato-carcinogenesis.

## 6. Patents

None.

## Figures and Tables

**Figure 1 viruses-16-01743-f001:**
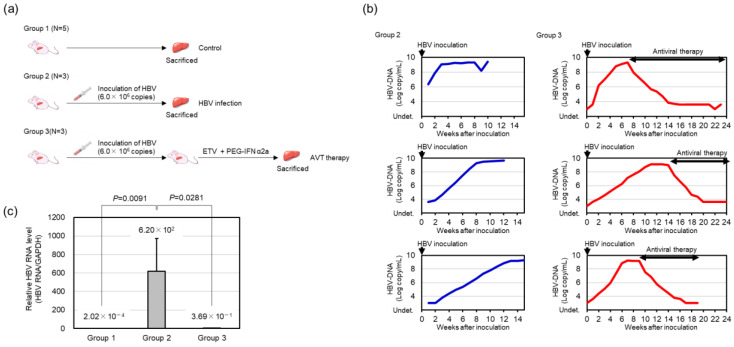
HBV viral load level in the livers of the HBV-infected uPA/SCID mice with humanized livers before and after antiviral therapy with entecavir and pegylated interferon. (**a**) Experimental schematic: Group 1, five non-infected mice were sacrificed to assess baseline gene expression; Group 2, three HBV-infected mice were inoculated with 6.0 × 10^6^ copies of HBV via the mouse tail vein, and sacrificed after the HBV DNA level had plateaued; Group 3, the remaining three HBV-infected mice were treated with entecavir plus PEG-IFNα2a until HBV DNA became undetectable. (**b**) Three HBV-infected mice in Group 2 were sacrificed after the HBV DNA levels had plateaued. Three HBV-infected mice in Group 3 were sacrificed after HBV DNA fell below detectable levels following AVT with entecavir plus pegylated interferon. (**c**) Intrahepatic HBV RNA levels (mean + SD) were measured by real-time PCR and were compared among the three groups. Statistical analysis was performed by an ANOVA test.

**Figure 2 viruses-16-01743-f002:**
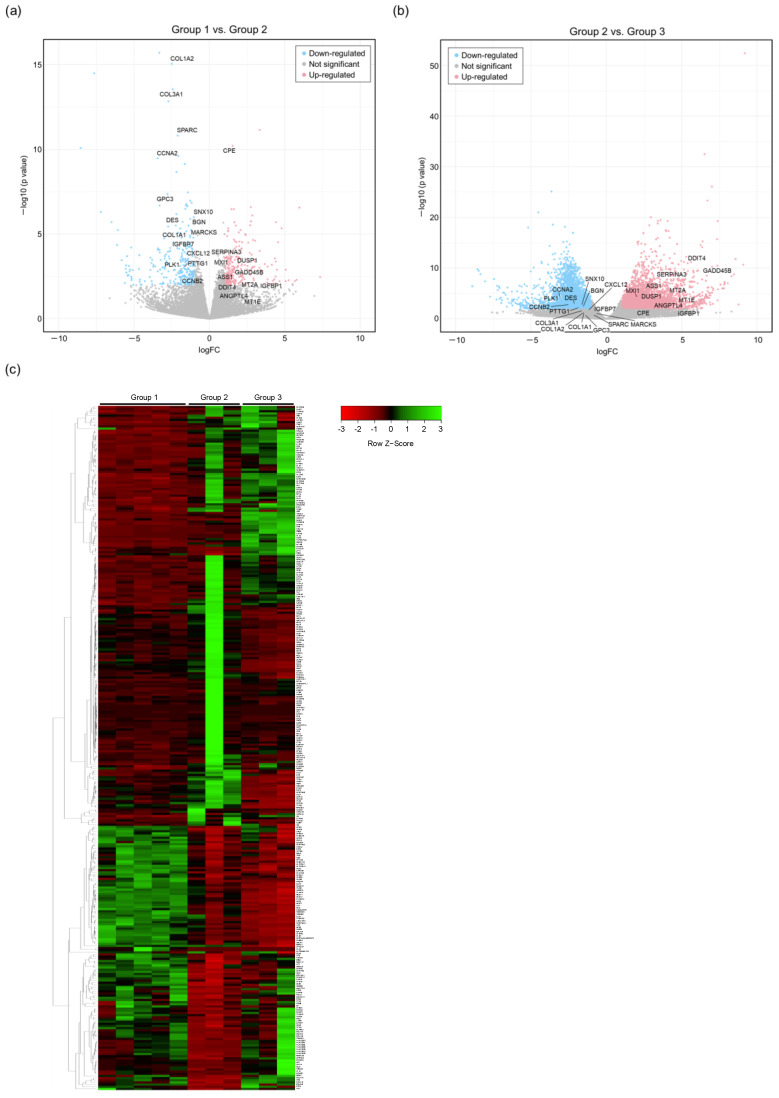
Altered gene expression profiles of intrahepatic gene expression between the HBV-infected and uninfected mice. (**a**,**b**) Alterations of all genes are shown by volcano plot; (**a**) Group 1 vs. Group 2; (**b**) Group 2 vs. Group 3. (**c**) Heatmap of differentially expressed genes (DEGs) between Group 1 (uninfected mice), Group 2 (HBV-infected mice), and Group 3 (HBV-infected mice with antiviral therapy) in the RNA-seq analysis are shown. Each column represents a mouse, and each line represents a DEG. Gene expression levels differed significantly among Groups 1, 2, and 3 (*p* value < 0.01). Genes are ranked based on z-score. Red, downregulated; green, upregulated.

**Figure 3 viruses-16-01743-f003:**
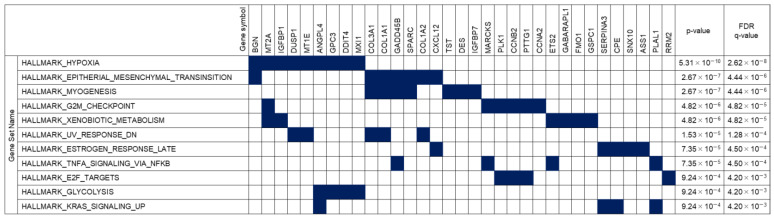
Pathways associated with the recovery of gene expression following antiviral therapy. A gene set enrichment analysis was performed using the set of genes for which expression recovered following AVT. Filled squares indicate the DEGs associated with Hallmark gene sets. Note: BGN, biglycan; MT2A, metallothionein 2A; IGFBP1, insulin-like growth factor-binding protein 1; DUSP1, dual-specificity protein phosphatase 1; MT1E, metallothionein 1E; ANGPTL4, angiopoietin-like 4; GPC3, glypican 3; DDIT4, DNA damage-inducible transcript 4; MXI1, MAX interactor 1; COL3A1, collagen type III alpha 1 chain; COL1A1, collagen type I alpha 1 chain; GADD45B, growth arrest and DNA damage-inducible protein (GADD) 45 beta; SPARC, secreted protein acidic and rich in cysteine; COL1A2, collagen type I alpha 2 chain; CXCL12, C-X-C motif chemokine ligand 12; TST, thiosulfate sulfurtransferase; DES, desmin; IGFBP7, insulin-like growth factor-binding protein 7; MARCKS, myristoylated alanine-rich protein kinase C substrate; PLK1, Polo-like kinase 1; CCNB2, cyclin B2; PTTG1, pituitary tumor-transforming gene 1; CCNA2, cyclin A2; ETS2, ETS proto-oncogene 2; GABARAP, GABA type A receptor-associated protein; FMO1, flavin-containing dimethylaniline monoxygenase 1; GSPT1, G1 to S phase transition 1; SERPINA3, serpin family A member 3; CPE, carboxypeptidase E; SNX10, sorting nexin 10; ASS1, argininosuccinate synthase 1; PLAGL1, PLAG1-like zinc finger 1; RRM2, ribonucleotide reductase regulatory subunit M2.

**Figure 4 viruses-16-01743-f004:**
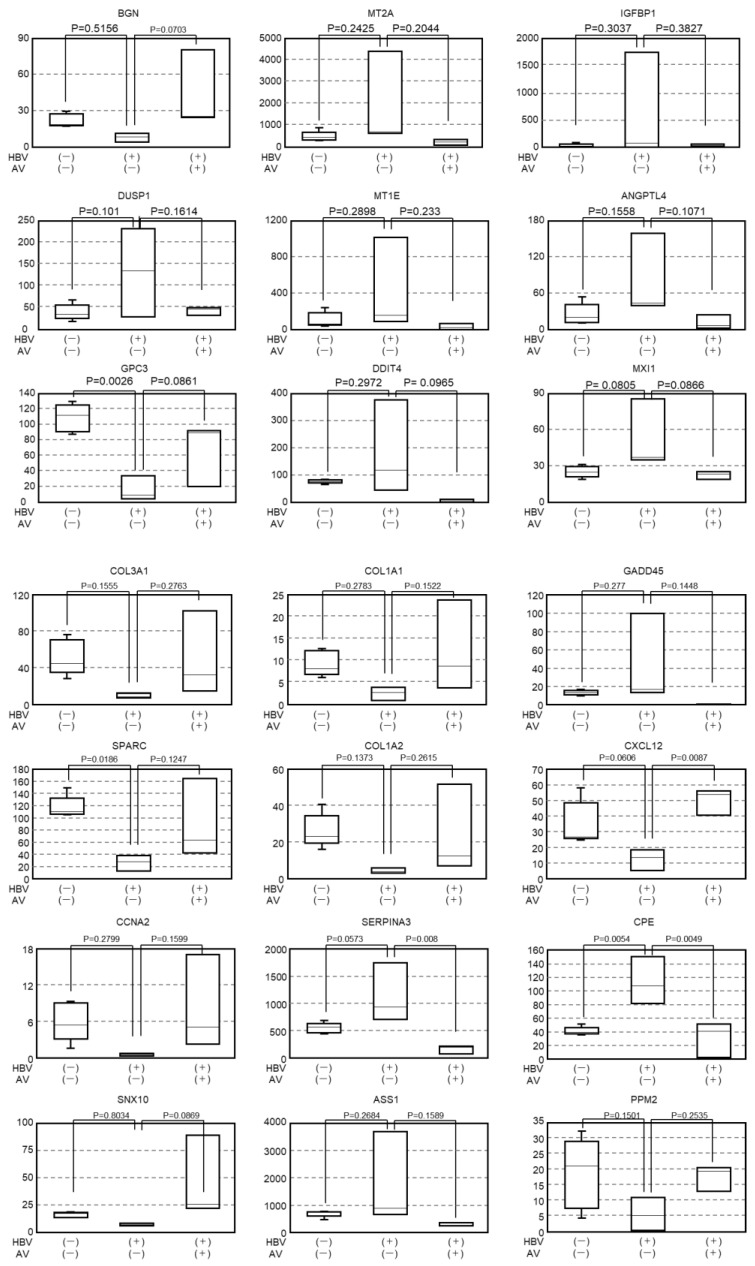
Expression of the genes associated with the recovered pathways. Dynamic changes in gene expression level in the non-infected mice, HBV-infected mice, and HBV-infected mice after antiviral therapy. Note: BGN, biglycan; MT2A, metallothionein 2A; IGFBP1, insulin-like growth factor-binding protein 1; DUSP1, dual-specificity protein phosphatase 1; MT1E, metallothionein 1E; ANGPTL4, angiopoietin-like 4; GPC3, glypican 3; DDIT4, DNA damage-inducible transcript 4; MXI1, MAX interactor 1; COL3A1, collagen type III alpha 1 chain; COL1A1, collagen type I alpha 1 chain; GADD45B, growth arrest and DNA damage-inducible protein (GADD) 45 beta; SPARC, secreted protein acidic and rich in cysteine; COL1A2, collagen type I alpha 2 chain; CXCL12, C-X-C motif chemokine ligand 12; TST, thiosulfate sulfurtransferase; DES, desmin; IGFBP7, insulin-like growth factor-binding protein 7; MARCKS, myristoylated alanine-rich protein kinase C substrate; PLK1, Polo-like kinase 1; CCNB2, cyclin B2; PTTG1, pituitary tumor-transforming gene 1; CCNA2, cyclin A2; ETS2, ETS proto-oncogene 2; GABARAP, GABA type A receptor-associated protein; FMO1, flavin-containing dimethylaniline monoxygenase 1; GSPT1, G1 to S phase transition 1; SERPINA3, serpin family A member 3; CPE, carboxypeptidase E; SNX10, sorting nexin 10; ASS1, argininosuccinate synthase 1; PLAGL1, PLAG1-like zinc finger 1; RRM2, ribonucleotide reductase regulatory subunit M2.

**Table 1 viruses-16-01743-t001:** Top 10 signaling pathways regulated by HBV infection.

No.	Pathway	Genes
1	Gonadotropin-releasing hormone receptor pathway (P06664)	HSPA1A, DUSP1, GNB5, HSPA1B, GATA4, and FST
2	Integrin signaling pathway (P00034)	COL3A1, RND1, COL1A1, and COL1A2
3	CCKR signaling map (P06959)	PLAU and CPE
4	Huntington disease (P00029)	CAPN6
5	p53 pathway (P00059)	CCNB1, RRM2, and GADD45B
6	T cell activation (P00053)	HLA-DQA1, CD74, FOS, HLA-DQA2, HLA-DQA1, and NFKBIA
7	Apoptosis signaling pathway (P00006)	HSPA1A, FOS, HSPA1B, and NFKBIA
8	Inflammation mediated by chemokine and cytokine signaling pathway (P00031)	COL6A3, JUND, COL14A1, and NFKBIA
9	Parkinson’s disease (P00049)	HSPA1A, HSPA1B, and SEPTIN4
10	Interleukin signaling pathway (P00036)	FOS and IRS2

Note: HSPA1A, Heat Shock Protein Family A Member 1A; DUSP1, dual-specificity protein phosphatase 1; GNB5, Guanine Nucleotide-Binding Protein Subunit Beta-5; HSPA1B, Heat Shock Protein Family A Member 1B; GATA4, GATA-Binding Protein 4; FST, Follistatin; COL3A1, collagen type III alpha 1 chain; RND1, Rho Family GTPase 1; COL1A1, collagen type I alpha 1 chain; COL1A2, collagen type I alpha 2 chain; PLAU, Plasminogen Activator, Urokinase; CPE, carboxypeptidase E; CAPN6, Calpain 6; CCNB1, Cyclin B1; RRM2, ribonucleotide reductase regulatory subunit M2.; GADD45B, growth arrest and DNA damage-inducible protein (GADD) 45 beta; HLA-DQA1, Major Histocompatibility Complex, Class II, DQ Alpha 1; CD74, Major Histocompatibility Complex, Class II, invariant chain; FOS, Fos Proto-Oncogene; HLA-DQA2, Major Histocompatibility Complex, Class II, DQ Alpha 2; HLA-DQA1, Major Histocompatibility Complex, Class II, DQ Alpha 1; NFKBIA, NFKB Inhibitor Alpha; HSPA1A, Heat Shock Protein Family A Member 1A; FOS, Fos Proto-Oncogene; HSPA1B, Heat Shock Protein Family A Member 1B; NFKBIA, NFKB Inhibitor Alpha; COL6A3, Collagen Type XIV Alpha 1 Chain; JUND, JunD Proto-Oncogene; COL14A1, Collagen Type XIV Alpha 1 Chain; NFKBIA, NFKB Inhibitor Alpha; HSPA1A, Heat Shock Protein Family A Member 1A; HSPA1B, Heat Shock Protein Family A Member 1B; SEPTIN4, Septin 4; FOS, Fos Proto-Oncogene; IRS2, Insulin Receptor Substrate 2.

## Data Availability

The original contributions presented in the study are included in the article, further inquiries can be directed to the corresponding authors.

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
