# Peer review of "Alteration of Gene Expression After Entecavir and Pegylated Interferon Therapy in HBV-Infected Chimeric Mouse Liver"

_viruses, 2024, doi:10.3390/v16111743_

Round 1
Reviewer 1 Report
Comments and Suggestions for Authors
In the abstract, please indicate that gene expression was evaluated using next-generation sequencing.
In section 2.5, please provide a detailed account of the methodology employed to differentiate human hepatocytes from mouse hepatocytes.
Please provide further details regarding the transcriptomic analysis. Please specify the bioinformatics tools utilized in this process. Please describe the criteria used to identify differentially expressed genes.
Figure 1c should indicate the statistical significance of the differences in mRNA levels.
Please construct volcano plots based on the RNA-seq data. The genes that exhibited the most dramatic changes in expression should be indicated on the plots. It would be advantageous to discuss the relevance of these genes to the action of the antiviral therapy.
It is recommended that the differential expression of the most relevant genes be verified using RT-PCR.
Author Response
Response to reviewers
Reviewer #1
- In the abstract, please indicate that gene expression was evaluated using next-generation sequencing.
Following to the reviewer’s comment, we indicated the use of next-generation sequencing in the Abstract (Line 15 -18).
- In section 2.5, please provide a detailed account of the methodology employed to differentiate human hepatocytes from mouse hepatocytes.
As the replacement rate of human hepatocytes can be estimated by serum human albumin level (Tateno C, et al. Am J Pathology, 2004), we were able to select human hepatocyte chimeric mice for which more than 90% of the mouse hepatocytes were replaced with human hepatocytes. Therefore, the obtained liver tissue overwhelmingly represented human hepatocytes. To improve the accuracy of gene expression analysis, we excluded reads that appeared to be from mRNA from mouse-derived cells. To clarify the methodology of gene expression analysis, we added the above sentences in the Methods (Line 88 – 89 and Line 146).
- Please provide further details regarding the transcriptomic analysis. Please specify the bioinformatics tools utilized in this process. Please describe the criteria used to identify differentially expressed genes.
Takara Bio used Expression Miner 2.0 software to analyze and visualize the RNA-Seq data. Unfortunately, the details of this internal software pipeline are not available to the public. Therefore, it is difficult to explain the methods in detail beyond this point (Line 145). We selected differentially expressed genes using the following criteria: the significance of the differences between Groups 1 and 2 was p < 0.01 with a ≥2-fold change in gene expression (Line 151).
- Figure 1c should indicate the statistical significance of the differences in mRNA levels.
We added P values in Fig. 1c.
- Please construct volcano plots based on the RNA-seq data. The genes that exhibited the most dramatic changes in expression should be indicated on the plots. It would be advantageous to discuss the relevance of these genes to the action of the antiviral therapy.
Thank you for the helpful comment. We added volcano plots (Fig. 2a) and discussed the relevance of these genes in the context of antiviral therapy.
- It is recommended that the differential expression of the most relevant genes be verified using RT-PCR.
We agree that RT-PCR results might bolster some of our findings, and we began performing RT-PCR to verify the RNA seq results, but we could not complete it due to the close deadline.
Reviewer 2 Report
Comments and Suggestions for Authors
The authors analyzed gene expression in HBV-infected human hepatocytes with or without AVT by NGS. The authors used humanized livers in uPA/SCID mouse model, which is quite unique and informative. Hepatitis B treatment remains an important human health issue and the results described in this manuscript will contribute a lot to this field especially hepato-carcinogenesis. Specific comments follow.
Major points:
1. Figure 1c: Please explain the error bar.
2. Figure 3: Please explain what does filled square mean.
3. Figure 4: Please add statistic differences in the graphs.
4. Please prepare a Table for abbreviation of gene symbols used in Figures 3 & 4.
5. Please combine the last paragraph in Discussion section (Line 295) to the paragraph one before (Line 268), as both paragraphs are talking about study limitations.
Minor points:
1. Line 128: Typo? “12” should be “11”?
2. Line 223: Please delete a space between “transplant” and “ed”.
Author Response
Response to reviewers
Reviewer #2
Major points:
- Figure 1c: Please explain the error bar.
We added an explanation of the error bar in the figure legends.
- Figure 3: Please explain what does filled square mean.
We indicated the DEGs belonging to the listed pathway and added an explanation about the meaning of the filled square in the figure legends.
- Figure 4: Please add statistic differences in the graphs.
We added P values in Figures 1c and 4.
- Please prepare a Table for abbreviation of gene symbols used in Figures 3 & 4.
We added abbreviations of the gene symbols in the legends of Figures 3 and 4.
- Please combine the last paragraph in Discussion section (Line 295) to the paragraph one before (Line 268), as both paragraphs are talking about study limitations.
Following the reviewer’s recommendation, we combined the paragraphs (Line 315 – 347).
Minor points:
- Line 128: Typo? “12” should be “11”?
Thank you for pointing out the error. We corrected it (Line 130).
- Line 223: Please delete a space between “transplant” and “ed”.
Thank you for pointing out this typo. We corrected it (Line 270).
Round 2
Reviewer 1 Report
Comments and Suggestions for Authors
Authors have improved the manuscript according to all the comments. I suggest that the revised manuscript be accepted for publication.